# FusionOcc: Multi-Modal Fusion for 3D Occupancy Prediction

## ABSTRACT

3D occupancy prediction (OCC) aims to estimate and predict the semantic occupancy state of the surrounding environment, which is crucial for scene understanding and reconstruction in the real world. However, existing methods for 3D OCC mainly rely on surround-view camera images, whose performance is still insufficient in some challenging scenarios, such as low-light conditions. To this end, we propose a new multi-modal fusion network for 3D occupancy prediction by fusing features of LiDAR point clouds and surround-view images, called FusionOcc. Our model fuses features of these two modals in 2D and 3D space, respectively. By integrating the depth information from point clouds, a cross-modal fusion module is designed to predict a 2D dense depth map, enabling an accurate depth estimation and a better transition of 2D image features into 3D space. In addition, features of voxelized point clouds are aligned and merged with image features converted by a view-transformer in 3D space. Experiments show that FusionOcc establishes the new state of the art on Occ3D-nuScenes dataset, achieving a mIoU score of 35.94% (without visibility mask) and 56.62% (with visibility mask), showing an average improvement of 3.42% compared to the best previous method. Our work provides a new baseline for further research in multi-modal fusion for 3D occupancy prediction.

## CCS CONCEPTS

• **Computing methodologies** → **Scene understanding**.

## KEYWORDS

3D occupancy prediction, Scene understanding, Cross-modal fusion, Depth estimation

## 1 INTRODUCTION

Perceiving and modeling the surrounding scenes accurately and comprehensively plays an important role in robotics navigation, autonomous driving and so on. Modern researches for 3D perception mainly focus on 3D object detection and semantic map reconstruction, where objects are described by 3D bounding boxes and generated maps are used for path planning.

While the 3D bounding box usually erases the geometric details of objects[3, 37], 3D occupancy prediction (OCC) offers a meticulous partitioning of the 3D space into semantically labeled voxels, which provides richer semantic and geometric expressiveness. 3D OCC is now widely used in the field of autonomous driving, providing drivers with detailed road conditions and a better driving experience.

Compared with 3D object detection [40], LiDAR segmentation [47], and semantic map reconstruction, estimating the detailed occupancy states and semantics of a scene is more complicated [2, 21]. Besides, the immense number of voxels brings new challenges to computation, necessitating the employment of techniques such as sparse convolution.

Some previous works have focused on vision-based occupancy prediction, aiming to avoid additional economic costs from LiDAR sensors. Typically, forward projection (LSS[33]) and backward projection (BEVFormer[18]) is widely used among their models, where 2D features of images are transformed into 3D features in BEV space by 2D-3D projection. The core of LSS is depth estimation, which is tough only using monocular images, especially for distant and small objects. Incorrect estimations of depth could lead to the wrong projection of 2D-3D, resulting in poor prediction[24]. Besides, encoding the image features directly into flattened BEV features is not feasible, causing the loss of geometry information.

Other works using LiDAR or fusing LiDAR with camera may easily introduce LiDAR-to-camera's distortion and camera-to-LiDAR's sparsity, which have been illustrated by BEVFusion[24]. Besides, geometry distortion by projecting LiDAR to camera becomes severer when the scene is dynamic [15], and point clouds are much more sparse when compared with dense voxels in OCC task. Thus, it is not feasible to model the occupancy state of the scene solely by temporally accumulating the point clouds.

In this paper, we propose a new network named FusionOcc to unify camera and LiDAR features in the shared 3D representation space for 3D occupancy prediction. The model fuses features of these two modals in 2D and 3D space, respectively. We design a cross-modal fusion module supervised by both sparse point clouds and image semantics, which provide an accurate depth estimation for the transition of 2D image features into 3D space. In addition, features of LiDAR point clouds are aligned and merged with image features converted by a view-transformer in 3D space. Finally, the fused features are fed into a specific occupancy head to predict the occupancy state of 3D space. Validated by extensive experiments, our model has great ability to reconstruct smaller objects and distant scenes. Besides, the model is less vulnerable to harsh weather conditions and insufficient lighting, achieving higher scores than all previous methods.

In summary, our main contributions are:

1. A new multi-modal fusion model is proposed for 3D occupancy prediction.

2. Our model, named FusionOcc, realizes a cross-modal fusion module that incorporates camera images and LiDAR point clouds to obtain an accurate depth estimation, enabling a better transition of 2D features into the 3D space.

3. FusionOcc achieves state-of-the-art performance on the OCC3D-nuScenes dataset and sets up a new baseline for 3D occupancy prediction based on multi-modal fusion.

## 2 RELATED WORK

### 2.1 Vision-based Occupancy Prediction

Occupancy Networks were originally proposed by Mescheder et al.[26, 32], focusing on continuously representing objects in 3D space. Recent variations in occupancy networks have shifted their focus to reconstructing 3D space and predicting voxel-level semantic information from image inputs. SurroundOcc[42] proposes a coarse-to-fine architecture for occupancy prediction. OccNet[39] exploits applying universal occupancy features to various downstream tasks and introduces the OpenOcc benchmark.

With the release of OCC-3d nuScenes[38] dataset, many excellent methods emerged in the CVPR 2023 occupancy challenge. FB-OCC[19] adopts a unified design that leverages both forward projection (represented by List-Splat-Shoot[33]) and backward projection (represented by BEVFormer[18]), promoting the benefits from each method with improved perception results while overcoming their limitations. UniOCC[31] approached the 3D occupancy task as a rendering problem, attempting to solve it using NeRF's[41] methodology. They also employed a Teacher-Student training approach to train the model.

### 2.2 Semantic Scene Completion

Another related task is Semantic Scene Completion (SSC)[8, 21, 43], and the main difference between SSC and 3D Occupancy Prediction (OCC) is that SSC usually deals with static scenes while OCC focus on dynamic ones. Though these two tasks differ a little, there are still many excellent works on SSC worthy of reference.

SSCNet[36] was the first to address SSC using both image and depth map data, utilizing a 3D convolutional network to generate occupancy and labels within a voxel grid. MonoScene[5] designed a unified camera-based approach for indoor and outdoor scenarios, leveraging line-of-sight projection and an innovative frustum proportion loss. OccDepth[27] utilizes stereo images and corresponding depth supervision. Other approaches incorporate the bird-eye-view (BEV) and temporal information for predicting 3D occupancy. This concept has been further expanded with the Tri-Perspective View[35].

Notably, LiDAR-based methods have shown superior performance compared to camera-based approaches in outdoor scenes. Christoph et al.[34] propose a scene segmentation model for scene completion. A global scene completion function is subsequently assembled from the localized function patches. S3cnet[7] utilizes a sparse convolution based neural network to fuse 2D and 3D point features. Apart from these methods, datasets such as SSCBench[16] and SemanticKITTI[21], providing annotated ground truth for SSC evaluation, facilitating a more comprehensive assessment of SSC methods.

### 2.3 Multi-modal Fusion

Recently, multi-modal fusion has aroused increased interest in the 3D perception and scene understanding community. Early approaches can be classified into proposal-level fusion and point-level fusion. Proposal-level fusion methods are object-centric and cannot trivially generalize to other tasks such as BEV map segmentation. Point-level fusion methods, on the other hand, usually paint image semantic features onto foreground LiDAR points and then perform LiDAR-based detection on the decorated point cloud inputs, such as Deep Continuous Fusion [20], DeepFusion[17] and so on.

Besides, BEVFusion[24] unifies multi-modal features in a shared bird's-eye view representation space for task-agnostic learning, where both geometric structure and semantic density are maintained. ObjectFusion[4] learns three kinds of modality-specific feature maps (i.e. voxel, BEV, and image features) from the LiDAR point cloud and its BEV projections for 3D object detection, where features of three modalities are further fused at the object level and finally fed into the detection head.

## 3 METHODOLOGY

The overview of our model is in Fig. 1. FusionOcc is composed of three main components: (1) A points branch that generates 3D point clouds features through a voxelization of point clouds and a voxel encoder. (2) A images branch that generates 3D image features through a cross-modal fusion of 2D semantics and depth maps, and a 2D-to-3D transformation. (3) An occupancy head that merges the 3D features of images and point clouds, then encodes the fused features to produce occupancy prediction. Specifically, the input of multiple frames of point clouds are merged to obtain relatively dense point clouds. After voxelization, sparse convolution is employed to generate the 3D features of point clouds. Along with the LiDAR branch, the camera branch utilizes images from camera and depth information from LiDAR. A cross-modal fusion module is adopted to fuse features of these two modals. Then, a view-transformation is adopted to obtain image features that are aligned with features of point clouds in common 3D space. Finally, features of two branches are merged and encoded by a 3D convolution network and then produce a prediction for the 3D occupancy state.

### 3.1 Points Branch

Given the multi-modal inputs of point clouds and image data, two modality-specific encoders extract point cloud features and image features respectively. Formally, for the points branch, the input point clouds consist of a set of points and each point is represented as a 5-dimensional vector $p = (x, y, z, i, r)$, where $x, y, z$ is the coordinates in 3D space, $i$ is the reflection intensity and $r$ is the ring index.

The voxel encoder encodes the input LiDAR point cloud. Due to the sparsity of the original point cloud, the voxel encoder, correspondingly, utilizes sparse convolution and sparse down-sampling module rather than common 3d convolution to reduce computation. We use VoxelNet[6] as our backbone. As the final occupancy task requires the voxel feature map to be consistent with the size and range of feature maps of images in the shared 3D space, we voxelize the LiDAR point cloud with 0.05m and apply a 8x down-sampling in the voxel encoder. The feature of point cloud in 3D space $B_l \in R^{C \times D \times H \times W}$ is generated by a dense convert operation followed sparse convolution.

### 3.2 Images Branch

Along with the points branch, the images branch is set to extract 3D features of multi-view images. The architecture of the camera

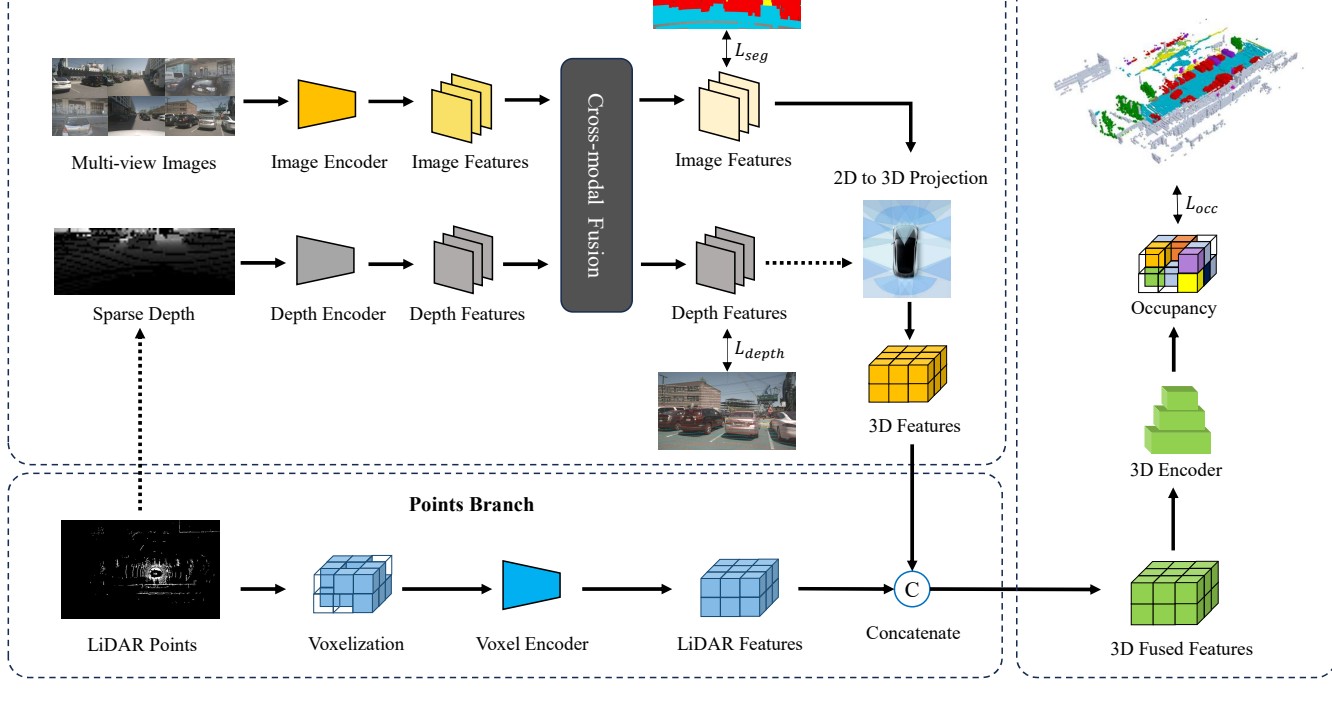

**Figure 1: The overall network architecture of this paper. The images branch and the points branch extract features of images and point cloud, then an occupancy head predicts the 3D occupancy state of the scene.**

branch is more complex compared with the LiDAR branch, and could be divided into three sub-parts.

**Image Encoder.** The input multi-view images of the branch represents a set of RGB images $R^{H \times W \times 3}$, where $H$ and $W$ is the height and width of the image. To exploit the power of multi-resolution features, the image encoder includes a backbone for high-level feature extraction and a neck for multi-resolution feature fusion. Besides, we utilize the temporal-fusion of multiple camera frames proposed by BEVDet4D[12]. Specifically, SwinTransformer[23] is utilized to extract 2D image feature maps and FPN-LSS[13] is adopted to fuse all feature maps of different scales into a single feature map. The module down-samples the feature to 1/16 input size.

**Cross-modal Fusion.** Modern methods unify multi-modal feature maps via BEV-based fusion where image features are projected into BEV space through camera-to-BEV transformation. However, the camera-to-BEV transformation hinges on pre-learned depth estimator to obtain the inherently ambiguous depth estimation of each pixel. Any inaccurate depth estimation can potentially result in spatial misalignment between the image feature maps and point feature maps. Besides, it is worth noting that semantic-level supervision from 2D images could enhance the prediction for 3D occupancy.

Based on that, a cross-modal fusion module is designed to unify images and sparse depth maps generated by corresponding point cloud at various levels. Dense depth map of RGB images is predicted by the module. Besides, a segmentation branch is adopted to predict

the 2D semantic masks alongside the depth prediction task and two branches' features are interacted with each other via a cross-attention mechanism. Details are shown in Fig. 2, the inputs of the cross-modal fusion module are the feature map of RGB images $F_c \in R^{H \times W \times C}$ and the features $F_d \in R^{H \times W \times C}$ of 2D depth map from point clouds. The 2D depth map is down-sampled x16 to keep the same resolution as image feature map and encoded using one-hot. Then, global average pooling is applied to $F_c$ and $F_d$ along the channel to obtain $\hat{F}_d$ and $\hat{F}_c \in R^C$. An MLP layer with a sigmoid function is followed to obtain $W_c$ and $W_d$:

$$W_d = \sigma(f_{mlp}(\hat{F}_d)) \tag{1}$$

Similar to SE-Net[11], we use $W_c$ and $W_d$ as the excitation module and a fused feature is obtained by element-wise multiplication along channel:

$$F_{d2c} = W_c * F_d$$
$$F_{c2d} = W_d * F_c \tag{2}$$

Followed by concatenating, fusing and splitting $F_{c2d}$ and $F_{d2c}$. Another average pooling along spatial level is applied to obtain $\tilde{F}_d$ and $\tilde{F}_c \in R^{H \times W}$. Then, an $1 \times 1$ convolution layer with Relu function is followed to obtain $Z_c$ and $Z_d$:

$$Z_d = Relu(Conv_{1 \times 1}(\tilde{F}_d)) \tag{3}$$

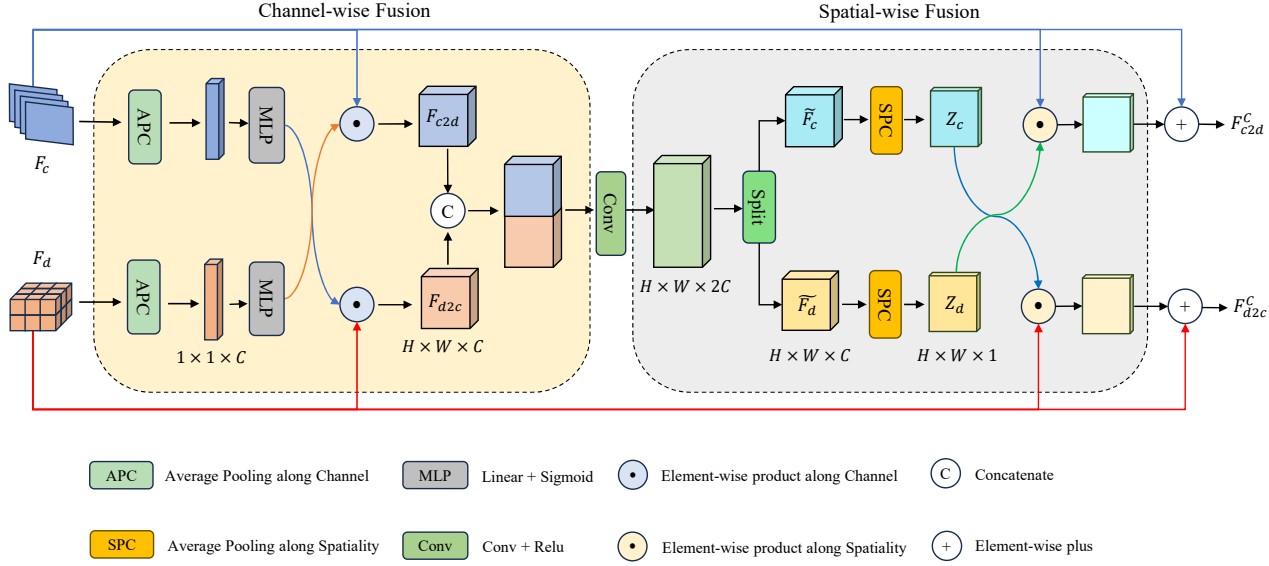

Figure 2: Details of the cross-modal fusion module. $F_c$ and $F_d$ represent the image features and depth features, respectively. $F_{c2d}^C$ and $F_{d2c}^C$ are the output of the module generated by channel-wise fusion(colored by yellow) and spatial-wise fusion(colored by green) of $F_c$ and $F_d$.

Likewise, we use $Z_c$ and $Z_d$ to obtain fused features by the element-wise multiplication and the original feature is added up to it:

$$F_{d2c}^C = \alpha Z_c * F_d + F_d$$
$$F_{c2d}^C = \alpha Z_d * F_c + F_c \qquad (4)$$

$\alpha$ is a hyper-parameter and the default value of them is set as 1 in Section 4. Features $F_{c2d}^C$ and $F_{d2c}^C$ are further concatenated for depth estimation and 2D semantic segmentation task, and the feature of segmentation branch is fed into the view transformation module to be projected into 3D space. The training strategy is designed to let the module to reconstruct the original depth maps generated by point clouds via randomly masked ones, then it could produce a dense depth map during the inference phase (see Fig. 3 ).

Also, dataset such as Occ3D-nuScenes[38] does not provide semantic segmentation labels for 2D images. To ensure consistency in semantic label categories and do not leverage other pre-trained models to generate pseudo labels, we use nuSences-lidarSeg[9] and 3D voxel labels to get the 2D semantic label via projection and up-sampling (see Fig. 4). We quantize the depth values of the point clouds to a fixed range, and classification loss is used to train the depth estimation and semantics segmentation branch. The total loss $L_{fuse}$ is the sum of depth loss $L_{depth}$ and segmentation loss $L_{seg}$:

$$L_{seg} = -\sum_{H,W} \left( y_{seg} log p_{seg} \right)$$
$$L_{depth} = -\sum_{H,W} \left( y_{depth} log p_{depth} \right)$$
$$L_{fuse} = L_{depth} + L_{seg} \qquad (5)$$

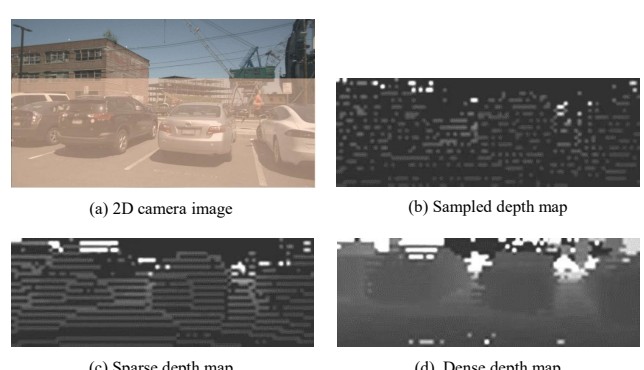

(a) 2D camera image     (b) Sampled depth map

(c) Sparse depth map     (d) Dense depth map

Figure 3: The strategy of generating a dense depth map of images. (a) The original 2D image (b) During the training phase, a randomly sampled depth map is fed to reconstruct the sparse depth map generated by point clouds. (c) During the inference phase, a dense depth map can be obtained when a sparse point clouds is provided as input.

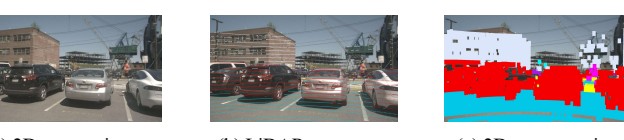

(a) 2D camera image    (b) LiDAR to camera    (c) 2D segmentation

Figure 4: Using the projection of point clouds to image to generate pseudo-labels for 2D image segmentation.(a) The original 2D image. (b) Projection of point clouds to 2D image. (c) Pseudo-labels generated by up-sampling.

where $H$ and $W$ are the height and width of the feature map which is down-sampled x16 compared to the input. $y$ denotes the label and $p$ denotes the prediction.

**2D-to-3D.** To obtain 3D image features, a view transformer [33] is utilized to transform the feature from image view to 3D space. The dense depth map predicted by cross-modal fusion module could assist in projecting the image feature into 3D space to obtain feature $B_c \in R^{C \times D \times H \times W}$.

## 3.3 Occupancy Head

Finally, 3D features of images and point clouds are merged along channel $C$ to produce a unified 3D feature. The unified feature is further encoded in 3D space and fed into an occupancy head to predict the 3D occupancy state. We utilize ResNet[10] with classical residual block to construct the module and combine the features with different resolutions by applying FPN-LSS[13]. A cross-entropy loss funtion $L_{occ}$ is used to train the occupancy prediction and the final loss is a weighted sum of $L_{occ}$ and $L_{fuse}$:

$$L_{total} = L_{occ} + \lambda * L_{fuse} \qquad (6)$$

where $\lambda$ is a hyper-parameter and is set as 0.1 in our model.

## 4 EXPERIMENTS

Our model is evaluated on the Occ3D-nuScenes[38] dataset. Occ3D-nuScenes is based on the nuScenes[3] dataset, which consists of large-scale multi-modal data collected from 6 surrounding cameras, 1 top LiDAR and 5 radars. The dataset is split into training/validation parts including 700/150 clips. The occupancy scope in the dataset is defined as -40m to 40m for X and Y axis, -1m to 5.4m for the Z axis, and 0.4m for the voxel size. Thus, the shape of the occupancy block is [200, 200, 16]. For 3D occupancy prediction task, the official evaluation metric uses mean Intersection over Union (mIoU) to evaluate models' performance:

$$mIoU = \frac{1}{N} \sum_{i=1}^{N} \frac{TP_i}{TP_i + FP_i + FN_i} \qquad (7)$$

where $TP_i$, $FP_i$, $FN_i$ represent the number of voxels that are predicted as true-positive, false-positive and false-negative of the class $i$, $N$ is the number of classes.

## 4.1 Implementation

We implement the proposed FunsionOcc in the PyTorch framework, based on the MMdetection3D and BEVDet[12] codebases. For the image encoder, we use the SwinTransformer[23] network as the backbone and FPN-LSS[13] to fuse multi-scale feature maps. Input images are cropped to remove invalid parts. The image backbone is pretrained on the nuImage[3] dataset for the 2D detection task. For the voxel encoder, we use VoxelNet[6] as the backbone to obtain point clouds features and down-sample the original point clouds by 8x. The voxel size of lidar is set as [0.05m, 0.05m, 0.05m], and the point clouds range is [-40m, -40m, -1m, 40m, 40m, 5.4m] in X, Y, and Z-axis, respectively.

During training, we utilize random flip, and random sample to augment the LiDAR data. Following common practice[1], we align the previous nine LiDAR sweeps into the current frame for

a denser point cloud. For images, we additionally use random flip, random rotation in and random resizing in to augment the images. AdamW[25] is used as the optimizer to train the model with a global batch size of 8. The initial learning rate is set as 1e-4 and is decayed with cosine annealing policy. For all experiments, we train our models for 20 epochs on 8 NVIDIA Tesla V100 GPUs.

## 4.2 Comparison with the State-of-the-arts

In Table 1, we compare FusionOcc with previous state-of-the-art methods on the validation split of Occ3D-nuScenes dataset. In order to make a fair comparison of the performance of these methods, we divide these methods into two groups based on whether they use the visibility mask or not, and the group where the visibility mask is used are marked with ∗.

Methods that do not use the visibility mask include OccFormer (camera-based method), VoxelNet (LiDAR-based method), BEV-Fusion (multi-sensor method) and so on. The input of VoxelNet is a fused point cloud of 8 adjacent frames and the output is encoded by the same encoder used in FusionOcc. 2DPASS-acc denotes the outstanding point cloud segmentation method, we accumulate the segmentation results of multi-frame point clouds to generate dense occupancy blocks. It is shown that although 2DPASS-acc has achieved good performance in point cloud segmentation tasks, the occupancy prediction based on the accumulated segmentation of multi-frame point clouds is relatively poor. Also, BEVFusion shares the same backbone, input size and OCC head with FusionOcc to keep fair. FusionOcc achieves the highest mIoU score of 35.94% among methods that do not use the visibility mask. As for methods that use the visibility mask including FB-OCC, UniOcc and so on, which have all achieved top scores in CVPR 2023 OCC competition, our model still outperforms them with a mIoU score of 56.62%.

## 4.3 Ablation Studies

**Effect of Cross-modal Fusion.** We employ a cross-modal fusion module to incorporate dense depth estimation and 2D image semantic segmentation. To testify its effectiveness, we conduct experiments in Table 2. Column 1 represents whether the model utilizes an additional learning branch where 4 different settings are adopted. Column 2 represents the fusion method used by the model.

From the table, it is observed that both the semantic branch and the depth branch have a positive impact on the accuracy of the model. The model with the two learning branches achieves the highest mIoU of 56.62%, which is a 2.56% improvement compared to the method without fusion and any additional learning branches. Besides, the designed cross-modal module has a more significant improvement (0.79%) in model accuracy compared to simply concatenating the features of the two modalities.

**Effect of Merging Multi-frames.** Since the point cloud at a single timestamp is sparse, we attempt to merge multiple frames to obtain a relatively dense point cloud. In Table 3, the mIoU, mAcc and F1 score of our model(with the visibility mask) is observed along different numbers of frames to be merged. From rows 1-4 of the table, it can be proved that the merging of point clouds of multiple frames is very effective. The performance of our model steadily improves as the number of frames increases, and the model

**Table 1: 3D occupancy prediction performance on the Occ3D-nuScenes validation set. ∗ denotes using the visibility mask. Our method achieves a higher performance than previous methods.**

| | Modality | others | barrier | bicycle | bus | car | cons.veh. | motorcycle | pedestrian | cons | trailer | truck | drv. surf. | other flat | sidewalk | terrain | manmade | vegetation | mIoU |
|---|---|---|---|---|---|---|---|---|---|---|---|---|---|---|---|---|---|---|---|
| MonoScene[5] | C | 1.8 | 7.2 | 4.3 | 4.9 | 9.4 | 5.7 | 4.0 | 3.0 | 5.9 | 4.5 | 7.2 | 14.9 | 6.3 | 7.9 | 7.4 | 1.0 | 7.7 | 6.1 |
| OccFormer[46] | C | 5.9 | 30.3 | 12.3 | 34.4 | 39.2 | 14.4 | 16.5 | 17.2 | 9.3 | 13.9 | 26.4 | 51.0 | 31.0 | 34.7 | 22.7 | 6.8 | 7.0 | 21.9 |
| BEVFormer[18] | C | 5.9 | 37.8 | 17.9 | 40.4 | 42.4 | 7.4 | 23.9 | 21.8 | 21.0 | 22.4 | 30.7 | 55.4 | 28.4 | 36.0 | 28.1 | 20.0 | 17.7 | 26.9 |
| TPVFormer[35] | C | 7.2 | 38.9 | 13.7 | 40.8 | 45.9 | 17.2 | 20.0 | 18.9 | 14.3 | 26.7 | 34.2 | 55.7 | 35.5 | 37.6 | 30.7 | 19.4 | 16.8 | 27.8 |
| CTF-Occ[38] | C | 8.1 | 39.3 | 20.6 | 38.3 | 42.2 | 16.9 | 24.5 | 22.7 | 21.1 | 23.0 | 31.1 | 53.3 | 33.8 | 38.0 | 33.2 | 20.8 | 18.0 | 28.5 |
| SparseOcc[22] | C | 10.6 | 39.2 | 20.2 | 32.9 | 43.3 | 19.4 | 23.8 | 23.4 | 29.3 | 21.4 | 29.3 | 67.7 | 36.3 | 44.6 | 36.4 | 22.0 | 21.9 | 30.9 |
| 2DPASS-acc[44] | L | - | 32.66 | 8.83 | 12.46 | 15.74 | 15.35 | 11.45 | 8.48 | 19.55 | 20.76 | 17.01 | 27.21 | 18.24 | 18.8 | 16.62 | 39.78 | 37.5 | 20.03 |
| VoxelNet[6] | L | 5.24 | 36.96 | 11.15 | 25.6 | 26.42 | 18.97 | 15.02 | 9.69 | 7.82 | 22.48 | 23.69 | 46.59 | 21.83 | 30.97 | 28.85 | 45.73 | 45.97 | 24.88 |
| BEVFusion[24] | C + L | 7.27 | 42.69 | 22.2 | 38.88 | 43.79 | 23.86 | 27.58 | 22.24 | 22.83 | 28.38 | 35.79 | 46.5 | 27.82 | 32.14 | 29.31 | 45.04 | 43.62 | 31.76 |
| **FusionOcc (Ours)** | C + L | 8.5 | 45.68 | 28.81 | 44.34 | 47.67 | 25.55 | 35.37 | 37.01 | 32.21 | 31.23 | 40.32 | 47.84 | 29.71 | 33.98 | 31.13 | 46.39 | 45.26 | **35.94** |
| FB-Occ*[19] | C | 14.30 | 49.71 | 30.0 | 46.62 | 51.54 | 29.3 | 29.13 | 29.35 | 30.48 | 34.97 | 39.36 | 83.07 | 47.16 | 55.62 | 59.88 | 44.89 | 39.58 | 42.06 |
| MiLO*[30] | C | - | - | - | - | - | - | - | - | - | - | - | - | - | - | - | - | - | 43.95 |
| UniOcc*[31] | C | - | - | - | - | - | - | - | - | - | - | - | - | - | - | - | - | - | 45.2 |
| BEVDet4d-occ*[12] | C | 12.13 | 49.58 | 24.98 | 51.94 | 54.36 | 27.77 | 27.9 | 28.91 | 27.21 | 36.38 | 42.2 | 82.32 | 43.34 | 54.59 | 57.88 | 48.56 | 43.5 | 41.97 |
| 2DPASS-acc*[44] | L | - | 39.67 | 11.09 | 16.65 | 19.72 | 21.41 | 13.01 | 9.88 | 23.03 | 29.31 | 22.39 | 34.0 | 23.23 | 23.49 | 21.79 | 50.92 | 45.79 | 25.34 |
| VoxelNet*[6] | L | 11.46 | 49.41 | 16.83 | 36.92 | 43.9 | 31.37 | 21.12 | 26.78 | 28.88 | 41.56 | 39.08 | 82.8 | 44.0 | 57.49 | 60.76 | 73.17 | 72.5 | 43.41 |
| BEVFusion*[24] | C + L | 16.2 | 61.94 | 39.34 | 58.22 | 62.51 | 38.13 | 41.62 | 46.68 | 47.65 | 50.55 | 52.72 | 85.72 | 49.38 | 60.68 | 64.25 | 71.72 | 70.24 | 53.97 |
| **FusionOcc* (Ours)** | C + L | 17.06 | 62.56 | 43.13 | 63.78 | 66.21 | 37.89 | 49.66 | 53.71 | 49.77 | 53.11 | 49.77 | 86.17 | 53.11 | 57.52 | 49.79 | 61.63 | 65.03 | **56.62** |

**Table 2: Ablation of cross-modal fusion. The first column indicates whether we use the semantics branch, depth branch or the combination of two to train the model. The second column indicates the method of fusing two modals.**

| Supervised | Fusion method | mIoU | mAcc | F1 |
|---|---|---|---|---|
| $L_{occ}$ | w/o fusion | 53.97 | 78.08 | 70.32 |
| $L_{occ} + L_{seg}$ | w/o fusion | 54.12 | 78.33 | 70.96 |
| $L_{occ} + L_{depth}$ | w/o fusion | 54.27 | 78.56 | 71.45 |
| | w/o fusion | 55.12 | 78.62 | 72.10 |
| $L_{total}$ | linear | 55.83 | 79.02 | 72.34 |
| | **cross-modal** | **56.62** | **79.30** | **72.47** |

**Table 3: Ablation of merging multiple frames. The first column C-Frames indicates the number of merged frames of cameras, while the second one L-Frames represents the number of merged frames of point clouds we use in the model.**

| C-Frames | L-Frames | mIoU | mAcc | F1 |
|---|---|---|---|---|
| | 1 | 51.43 | 71.80 | 68.31 |
| 2 | 2 | 53.48 | 78.71 | 70.02 |
| | 4 | 55.87 | 78.82 | 71.80 |
| | **8** | **56.62** | **79.30** | **72.47** |
| 1 | 8 | 55.32 | 78.76 | 71.72 |
| 3 | 8 | 56.16 | 78.71 | 72.13 |
| 4 | 8 | 56.68 | 79.12 | 72.49 |

**Table 4: Parameter amounts comparison of different methods. The original resolution of input images is all set as 1600 × 900. ∗ means using the visibility mask.**

| Method | Modality | Params(M) | mIoU(%) |
|---|---|---|---|
| InverMatrixVT3D[28] | C | 67.18 | 27.94 |
| SurroundOcc[42] | C | 180.51 | 31.49 |
| OccFusion[29] | C+L | 92.71 | 33.24 |
| | C+L+R | 114.97 | 34.35 |
| **FusionOcc** | C+L | 111.70 | **35.94** |
| FB-Occ*[19] | C | 67.8 | 42.06 |
| FB-Occ(large)* | C | 1200.0 | 52.79 |
| **FusionOcc(tiny)*** | C+L | 67.31 | **52.78** |
| **FusionOcc*** | C+L | 111.70 | **56.62** |

achieves the highest performance and obtains a 5.21% improvement on mIoU when the number of fused frames is 8.

From rows 5-6 of the table, the number of camera frames fused in the model is set as 1, 3 and 4. The result shows that even if more image features from previous timestamps are fused, there will not

be a significant change in the performance of the model (less than 1% point fluctuation).

## 4.4 Parameter Amounts Analysis

In addition, the number of parameters of our model is compared. Results are still divided into two groups (with and without visibility mask) for a fair comparison. We can see that in Tab. 4, even with a relatively small model size, FusionOcc can achieve a higher score on mIoU. To lighten the model, we adopt MobileSAM[45] as our backbone in the image branch(marked as **tiny**). MobileSAM is the ultra-lightweight version of SAM(Segment Anything Model)[14], which has been pre-trained on various open-domain datasets. As shown in the table, the model's size could be reduced to half without much accuracy loss.

## 4.5 Robustness Analysis

**Performance in Different Scenarios.** The accuracy of scene understanding and modeling is often greatly influenced by environmental conditions, such as rainy days and at night. Multi-modal fusion helps to improve the model's robustness in different scenarios. As seen in Fig. 5, our model FusionOcc achieves the top scores in three different scenarios. Even in rainy days when LiDAR is likely to be interfered, the accuracy of our model remains relatively stable with little change.

**Performance Varies with Different Perception Distances.** Multi-modal fusion not only improves the final model's robustness to different illumination and weather conditions but also extends the model's perception range. We take the vehicle as the center and define $R$ as the distance from the center. By setting different $R = [15m, 20m, 25m, 30m, 35m, 40m]$ (with the maximum value of R is 40), we study the performances of different models and modal fusion methods under different perception ranges.

As seen in Fig. 6, by fusing modals of cameras and LiDAR point clouds, the model's ability to perceive objects at extended distances is significantly improved. This phenomenon can be attributed to LiDAR data having a longer detection range and providing the modal of images with accurate depth information. Although features of point clouds are sparse, they can be combined to nearby features from other modalities, thereby enhancing the ability of the final merged feature to model the 3D scene.

**Performance Varies with Random Cameras Loss.** We demonstrate the robustness of our proposed modal against inferior camera conditions on the dataset. As shown in Fig. 7, we can see that even with a camera input lost, the model could still achieve a 53.66% mIoU score. With 3 camera inputs lost, the model achieves a 47.40% mIoU score. As expected, when the number of lost cameras increases, the gap between the performance of models that uses multi-modal fusion and single visual modality gradually widens. The reason is that when information from one modality is lost, the other modality can compensate for it.

## 4.6 Qualitative Results

In addition, we visualize the occupancy prediction of FusionOcc compared with VoxelNet, BEVdet4d-occ, and BEVFusion (see Fig 8). BEVFusion is a multi-sensor fusion method designed for the 3D object detection task and map segmentation, we modify it to be suitable for occupancy prediction task based on its architecture. VoxelNet and BEVdet4d-occ are chosen to represent the OCC model that only uses LiDAR or cameras.

In Fig. 8, three different scenarios are chosen (daytime, rainy day, and nighttime from the top to the bottom). In the daytime scene, we can see that BEVFusion misclassified the road surface and fence in the distance, while our model still predicts them correctly. In the rainy day scene, VoxelNet which uses only LiDAR as the input is the worst, and the prediction is badly influenced by motion blur of the vehicle. FusionOcc is the best compared to BEVDet4d-occ and BEVFusion, proved by its modeling of fine fences and bushes in close proximity. In the nighttime scene, models misclassify small subsections of the scene while our FusionOcc is still the one closest to the GT.

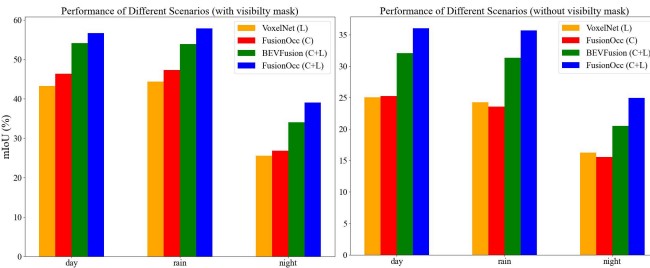

**Figure 5: Models' performances of different scenarios. Models are trained with visibility mask (left) and without visibility mask (right), respectively. Better viewed when zoomed in.**

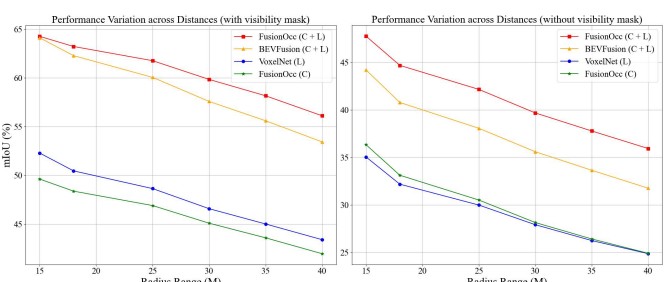

**Figure 6: Models' performance variation across distances. Models are trained with visibility mask (left) and without visibility mask (right), respectively. Better viewed when zoomed in.**

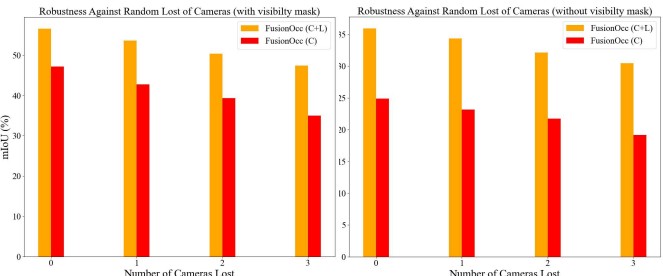

**Figure 7: Ablating the impact of malfunction of cameras. The model was evaluated in scenarios involving the random loss of one, two, or three camera data. Models are trained with visibility mask (left) and without visibility mask (right), respectively. Better viewed when zoomed in.**

## 5 CONCLUSION

This paper proposes a new 3D occupancy network, named FusionOcc, which is based on the fusion of camera and LiDAR features. FusionOcc realizes a cross-modal fusion module that incorporates images and point clouds to obtain an accurate depth estimation, enabling a better transition of 2D features into 3D space. Additionally, a new baseline for 3D occupancy prediction based on multi-modal fusion is established. Experiments show that FusionOcc achieves the state-of-the-art performance on Occ3D-nuScenes for mIoU. We

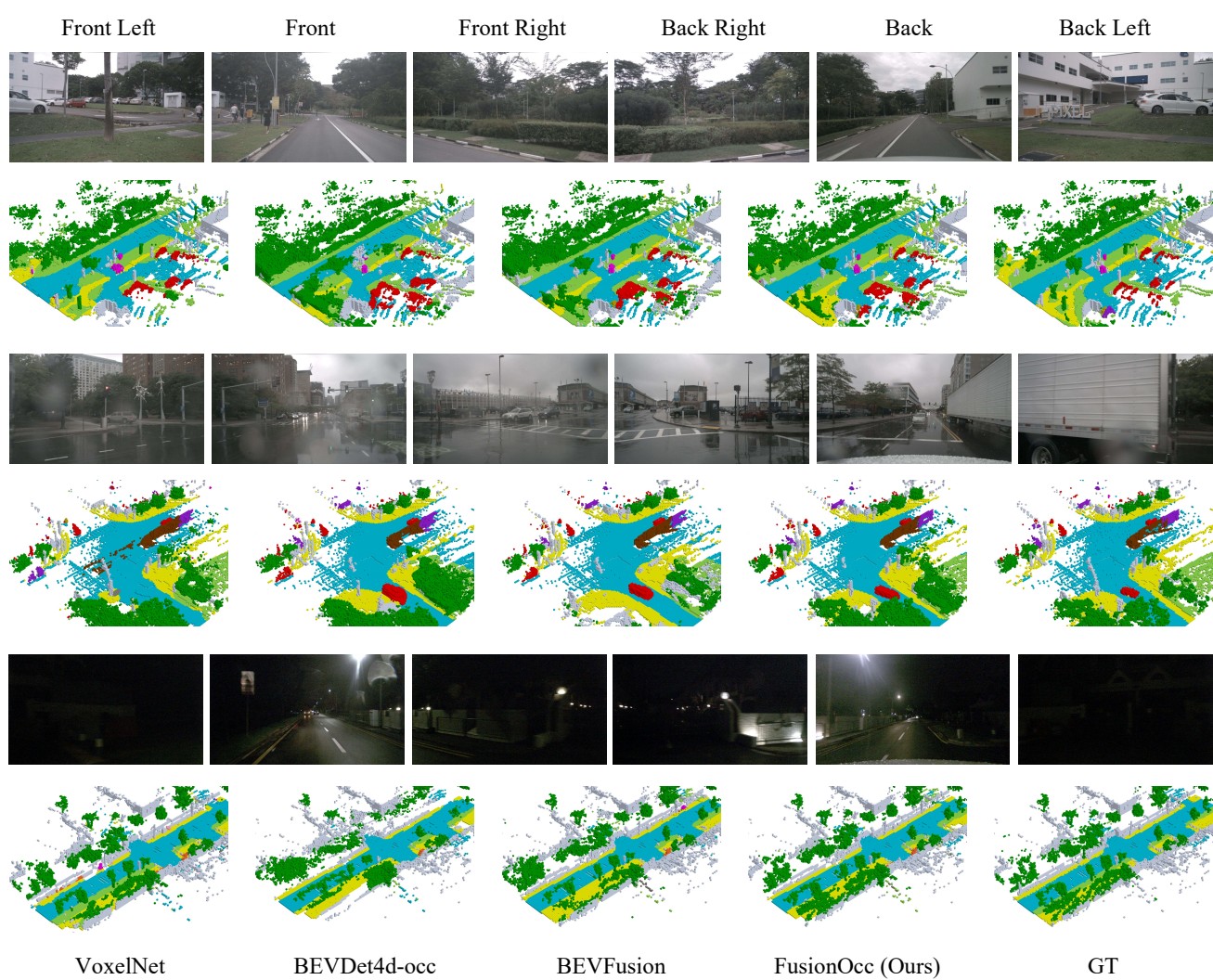

**Figure 8: Qualitative results of occupancy prediction on the validation set of Occ3D-nuScenes. Three different scenarios are chosen (daytime, rainy day, and nighttime from the top to the bottom), and our model achieved the best results in all of them.**

hope this result will attract more attention to the multi-task multi-modal fusion for 3D occupancy prediction.

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
