# OpenReview forum: "FusionOcc: Multi-Modal Fusion for 3D Occupancy Prediction"
_acmmm.org/ACMMM/2024/Conference — MM2024 Poster_

### Official Review · Reviewer_m4kz · 2024-05-25

**Rating:** 4
**Confidence:** 3

**Summary:**

This paper proposes a new 3D occupancy network, named FusionOcc, which is based on the fusion of camera and LiDAR features.
FusionOcc realizes a cross-modal fusion module that incorporates images and point clouds to obtain an accurate depth estimation, enabling a better transition of 2D features into 3D space. Additionally, a new baseline for 3D occupancy prediction based on multi-modal
fusion is established. Experiments show that FusionOcc achieves the state-of-the-art performance on Occ3D-nuScenes for mIoU.

**Strengths:**

1. This paper proposes a new multi-modal fusion model is proposed for 3D occupancy prediction FusionOcc.
2. FusionOcc realizes a cross-modal fusion module that incorporates camera images and LiDAR point clouds to obtain an accurate depth estimation, enabling a better transition of 2D features into the 3D space.
3. FusionOcc achieves state-of-the-art performance on the OCC3DnuScenes dataset and sets up a new baseline for 3D occupancy prediction based on multi-modal fusion

**Limitations:**

1.In fact, FusionOCC is BEVFusion combined with SENet for occupancy prediction. Please specifically explain the differences between them and the advantages of FusionOCC

2.In Figure 1, FusionOCC uses a weakly supervised method (similar to MAE) to supervise the training of the depth prediction module, aiming to use sparse or dense depth maps to obtain accurate 3D features. However, this method of obtaining depth is inaccurate and can lead to errors between the generated 3D features and the voxel features generated by LiDAR. Does this inaccuracy significantly impact occupancy prediction?

3.In Table 1, it is unreasonable to compare FusionOCC, which uses multimodal data, with other occupancy predictors that use only single modalities. It is recommended to compare FusionOCC with more methods that also use multimodal data

4.In Figure 8, the visualization shows the prediction results of FusionOCC and other methods. It is recommended to highlight the advantages of FusionOCC in the predictions. Currently, it is difficult to see the benefits of FusionOCC from the paper.

**Suitability:**

2

---

### Official Review · Reviewer_GT5h · 2024-05-25

**Rating:** 4
**Confidence:** 3

**Summary:**

This paper aims to integrate the features of LiDAR point clouds and surround-view images for 3D occupancy prediction task.
The main contributions include a cross-modal fusion module that combines camera iamges with LiDAR point clouds.
It achieves the state-of-the-art performance on the OCC3D-nuScenes dataset.

**Strengths:**

The proposed multi-modal fusion model incorporates camera images and LiDAR point clouds with a cross-modal fusion module.
And obtain accurate depth estimation to realize better transformation of 2D features into 3D space.
The method achieves the state-of-the-art performance on the OCC3D-nuScenes dataset.

**Limitations:**

1. The proposed method actually uses multiple frames of LiDAR point clouds and camera images, but in Figure 1, only one frame is displayed, and the sparse depth only has one view. It is better to display multi-frames of data and multi-view sparse depth.
2. In line 471, the authors claim that the dense depth map could assist in projecting the image feature into 3D space. Does this mean the dense depth can be used or not used in the 2D to 3D projection process. What is the impact of using dense depth on the final results?
3. The proposed method used 2 frames of camera images and 8 frames of LiDAR point clouds. It is better to point out the number of frames used by the comparison methods in Table 1. Because the number of frames used has a significant impact on the mIoU results.
4. In line 521, nine previous LiDAR sweeps are aligned into the current frame, which means that 10 frames of LiDAR point clouds are used during training, but according to Table 1 and Table 3, only 8 frames are used.
5. In line 725, camera input lost means that the network input does not include the camera image and corresponding sparse depth, or just set the RGB value of the camera image to a specific value.

**Suitability:**

3

---

### Official Review · Reviewer_nyt8 · 2024-05-27

**Rating:** 3
**Confidence:** 3

**Summary:**

This paper proposes a network for fusing LiDAR features and image features for 3D occupation prediction. The key idea is to use LiDAR point clouds to get depth information and generate depth images. And depth images are first fused with color images by a cross-modal fusion module and the fused images features are further fused with LiDAR features. Occupancy prediction loss, image segmentation loss, and depth prediction loss are jointly used to supervise learning. The proposed method achieves SOTA results.

**Strengths:**

1. Writing and the motivation is clear and reasonable.
2. The proposed method achieves SOTA results.

**Limitations:**

1. The key contribution of the proposed method is multi-modal fusion. However, the paper primarily demonstrates the fusion process without providing substantial insights into the design of the cross-modal fusion module. Consequently, the new knowledge introduced in this paper is somewhat limited.
2. From Table 2, the use of the cross-modal fusion module shows only marginal improvement over simple linear layers. This raises doubts about the effectiveness and necessity of the cross-modal fusion module, which is purportedly the most significant contribution of the paper.

**Suitability:**

3

---

### Official Review · Reviewer_a78J · 2024-05-31

**Rating:** 4
**Confidence:** 2

**Summary:**

This paper proposes a new multi-modal fusion network for 3D occ prediction by fusing features of LiDAR point clouds and surroundview images. By integrating the depth information from point clouds, a cross-modal fusion module is designed to predict a 2D dense depth map, enabling an accurate depth estimation and a better transition of 2D image features into 3D space. In addition, features of voxelized point clouds are aligned and merged with image features converted by a view-transformer in 3D space. Experimental results show its effectiveness and superiority among these SOTAs in occ prediction.

**Strengths:**

1. Promise performance in occ prediction. The proposed model overpasses the current SOTAs.
2. Easy to understand and interesting video in supplementary material.
3. Simple but effective motivation for integrating LiDAR point clouds for occ prediction.
4. Sufficient about proposed techniques.

**Limitations:**

My rating is borderline and will be adjusted based on the author's answer.

1. Some current works about converting 2D images into 3D voxels should be discussed and compared, especially in the introduction and experiments like PanoOcc, Occformer or RenderOcc. Row 69-79 can't not accurately summarize the latest progress of the current occ model by utilizing the deformable attention (PanoOcc or TPVFormer), especially TPVFormer designs TPV representation for balancing the  geometry information and memory cost.

2. Poor writing skills. What's the meaning of the "forward/backward projection" like row 71-72 ?  What's the difference between the previous works by utilizing Lidar-camera fusion and FusionOcc? How does the proposed model overcome the LiDAR-to-camera’s distortion problem?

3. Massive  logic in methods. What's the depth encoder?  How to exact the sparse depth from the LiDAR points?
"Dense depth map of RGB images is predicted by the module." What's "the module" ? Too many questions about the how to utilize depth information.

4. Unclear model setting and unfair compare. Mentioned that the proposed model utilizes the SwinT as backbone (MobileSAM) and temporal-fusion tricks (how to use it?), I can't see any ablation study about this, espeically some works utilize the R101-DCN as backbone like RenderOcc or BEVFormer. I think this work may mainly develop on the Bevdet codebase. For accurate evaluating these methods, authors should give more clear explanation about the model setting.

5. Authors should add more ablation experiments about $\alpha$ in Eq.4.

**Suitability:**

3

---

### Meta-Review · Area_Chair_WjX8 · 2024-06-27

**Recommendation:** Accept (Poster)
**Confidence:** 5

**Metareview:**

3D occupancy prediction is an important problem, and the authors propose a multi-modal fusion method that achieves encouraging performance. All the reviewers agree to accept this manuscript. The authors should carefully address the concerns in the final version, such as the writing of the paper, ablation studies, and some experimental comparisons.